# Development of a 3D Vascular Network Visualization Platform for One-Dimensional Hemodynamic Simulation

**DOI:** 10.3390/bioengineering11040313

**Published:** 2024-03-26

**Authors:** Yan Chen, Masaharu Kobayashi, Changyoung Yuhn, Marie Oshima

**Affiliations:** 1Graduate School of Interdisciplinary Information Studies, The University of Tokyo, 7-3-1 Hongo, Bunkyo-ku, Tokyo 113-0033, Japan; chenyan@iis.u-tokyo.ac.jp; 2Institute of Industrial Science, The University of Tokyo, 4-6-1 Komaba, Meguro-ku, Tokyo 153-8505, Japan; mkoba@iis.u-tokyo.ac.jp; 3Department of Mechanical Engineering, The University of Tokyo, 7-3-1 Hongo, Bunkyo-ku, Tokyo 113-0033, Japan; yuhn@iis.u-tokyo.ac.jp; 4Interfaculty Initiative in Information Studies, The University of Tokyo, 7-3-1 Hongo, Bunkyo-ku, Tokyo 113-0033, Japan

**Keywords:** blood flow simulation, data visualization, one-dimensional model, patient-specific, medical image

## Abstract

Recent advancements in computational performance and medical simulation technology have made significant strides, particularly in predictive diagnosis. This study focuses on the blood flow simulation reduced-order models, which provide swift and cost-effective solutions for complex vascular systems, positioning them as practical alternatives to 3D simulations in resource-limited medical settings. The paper introduces a visualization platform for patient-specific and image-based 1D–0D simulations. This platform covers the entire workflow, from modeling to dynamic 3D visualization of simulation results. Two case studies on, respectively, carotid stenosis and arterial remodeling demonstrate its utility in blood flow simulation applications.

## 1. Introduction

The advancement of computational fluid dynamics (CFD) has been pivotal in enhancing our understanding of hemodynamics. Among the various computational approaches, Reduced Order Models (ROMs) have emerged as a cost-effective alternative to high-dimensional numerical simulations. One of the instances for ROM is the one-dimensional (1D) model [1]. Reduced-order simulation stands out for its pragmatic balance between computational expense and result fidelity. Even in the era of machine learning and surrogate models, ROMs retain their applicability, anchored in their robust physics and mathematics foundation. This enables swift and manual customization to new demands, bypassing the necessity for extensive retraining or data acquisition. A comparative analysis of the different computational approaches, including 3D models, ROMs, and machine learning methods, is summarized in Table 1, highlighting these discrepancies in detail.

The advent of one-dimensional (1D) models has heralded significant advancements in the simulation of hemodynamic processes within the arterial system. Initially proposed in the late 1960s [2], these models have undergone extensive evolution over the past five decades. The 1D models, grounded in the seminal work by Hughes [1], have been instrumental in analyzing the propagation of pressure and flow waves in the arterial tree. Subsequent iterations have spanned from modeling partial arterial segments to comprehensive systemic networks [3,4,5,6,7,8,9,10,11]. These models have not only simulated generalized physiological conditions but also been refined to address the specificities of localized vascular networks, including cerebral and coronary arteries, displaying promising consistency with empirical patient data [12,13].

In clinical contexts, the adaptability of 1D models is paramount. Customization is achievable via modular integration or the assimilation of patient-specific data, enabling the simulation of particular clinical scenarios. The models have been enhanced by integrating the principles of arterial stenosis as formulated by Young [14], and through modifications to reflect individual peripheral resistance, as seen in the models by Liang [15,16] and the patient-specific simulations by Zhang et al. [17], which utilized SPECT data to calibrate vascular resistance. These advancements facilitate precise assessments of arterial flow dynamics, essential for diagnosing conditions and preventing complications such as Cerebral Hyperperfusion Syndrome (CHS). Furthermore, the capacity for these models to replicate the arterial remodeling in pathological states has been validated, as evidenced by the simulations of carotid artery stenosis [18] and abdominal artery stenosis [19]. Recent research has pivoted towards the quantification of uncertainty within these simulations [20], aiming to bolster the fidelity of 1D–0D models in mirroring the in vivo circulation and providing actionable insights to healthcare practitioners.

Previous studies have demonstrated a good consistency between 3D and 1D arterial blood flow simulations [21,22,23]. Xiao et al. [23] mentioned that when the flow is predominantly unidirectional and there are no abrupt changes in the cross-sectional area, one can expect a good consistency between the two modeling techniques, but in more complex configurations, such as in curved vessels and areas with significant height variations, larger discrepancies are anticipated. In these instances, introducing additional empirical models can help account for pressure losses in situations involving vascular curvature, narrowing, aneurysms, and other conditions. On the other hand, compared to 3D models, 1D models typically possess far fewer degrees of freedom which requires reasonable computational resources. This allows 1D models to be executed on a personal laptop within a matter of minutes.

1D simulation yields limited-fidelity results, which is largely attributed to their reliance on overly simplified models operating under idealized conditions. This simplification results in the loss of spatial detail. Furthermore, 1D simulations do not adequately capture the complexity of fluid dynamics in the circulation system. This results in that they are frequently employed to set conditions for high-dimensional simulations, drawing insights from their lower-dimensional counterparts. To harness the strengths of 1D simulation effectively, integrating user interaction is a key strategy. Specifically, by enabling users to examine and interpret ROM outputs, valuable insights can be gleaned. This is especially beneficial in medical settings, where it helps in preliminary decision making. Additionally, for hemodynamics research, user interaction with 1D simulations can be instrumental in identifying specific areas within blood vessels that warrant detailed 3D simulations. This targeted approach not only streamlines research efforts but also significantly reduces computational costs by focusing on critical regions rather than conducting extensive high-fidelity simulations across the entire vascular network.

In this study, we proposed a platform centered around the blood flow simulation 1D model proposed and improved by [15,16,17], equipped with a compact visualization module. This platform encompasses a complete suite of medical image-based vasculature modeling, simulation input/output, and result visualization. It is aimed at supporting hemodynamics research. The paper discusses the platform’s requirements, introduces its architecture, and presents two case study instances.

## 2. Requirements

The platform is designed to meet the following requirements:Medical Image Input/Output and Processing: This involves the processing of medical images, primarily focusing on segmentation to extract vascular regions from the image.Centerline Extraction: The platform should be able to obtain centerlines from extracted vascular regions, encompassing two key aspects: accurately capturing the vasculature’s topological structure and computing geometric parameters of the vessels, such as radius and length.Centerline Denoising: With the extracted centerlines, the platform should be able to remove noise inherent in the imaging data, generating smooth numerical data for simulation.Simulation Results Integration: This includes reading the simulation outputs and mapping these results onto the corresponding centerlines, ensuring a coherent relationship between the simulated data and the anatomical structures.Simulation Results Visualization: This involves generating surface representations of vessel lumens and displaying the simulation results on these surfaces, enabling clear and detailed visualization of hemodynamic parameters.

## 3. 1D–0D Computational Model

In this work, we utilized a 1D–0D entire circulation model as a case study to demonstrate the applicability of our hemodynamic visualization platform. In this section, we briefly introduce the characteristics of this simulation model. The introduction will focus on three aspects: (1) the input and output required by the model, (2) the calculation process of the model, and (3) the application of the model.

Blood flow simulation based on computational fluid dynamics (CFD) advances the field by providing a quantitative index for risk assessment, which facilitates minimally invasive approaches and aids in determining treatment strategies for revascularization. To apply the CFD-based blood flow simulation to clinical practice, it is essential to obtain results within a relatively short time. Typical three-dimensional analyses require extensive computational resources, especially when considering a wide range of blood vessels to determine local flow distribution. The considerable demands on computational capacity and time render these analyses impractical for rapid clinical application. Conversely, low-cost blood flow simulations using reduced-order models have been developed as diagnostic support tools. Liang [15,16] proposed a 1D–0D cardiovascular circulation model, where the 1D model computes the deformation of the vessel wall and the accompanying pulse wave propagation by applying the surface integral over the cross-sectional area of the governing equation. The 0D model simulates the mechanical characteristics of blood vessels using analogies with electric circuits. The two models are coupled to simulate blood flow through the entire circulatory system.

In this model, major arteries are represented as one-dimensional (1D) axisymmetric tubes, while peripheral blood vessels, veins, and the heart are modeled as zero-dimensional (0D) circuit elements. These 0D elements are coupled at the beginning and end of the 1D model, forming a closed loop, as illustrated in Figure 1. The 1D model governs the major arteries, including the ascending aorta, and the circulations of the upper (Circle of Willis, external carotid arteries, and upper limbs) and lower body (abdominal and lower limbs). The 1D components connect to the 0D components at the terminal arteries/arterioles of each circulation. Meanwhile, veins converge at the heart’s right atrium and ventricle, with blood flowing through the pulmonary circulation before being pumped into the ascending aorta from the left atrium and ventricle, thus re-entering the 1D regime and initiating a new cardiac cycle.

The 1D governing equations are obtained from the continuity and longitudinal momentum equations of the Navier–Stokes Equations [7,9]:(1)$$\frac{\partial A}{\partial t}+\frac{\partial Q}{\partial z}=0$$
(2)$$\frac{\partial Q}{\partial t}+\frac{\partial }{\partial z}\left(\frac{Q^{2}}{A}\right)+\frac{A}{\rho }\frac{\partial P}{\partial z}+K_{R}\frac{Q}{A}=0$$
where the unknowns *A*, *Q*, and *P* denote the cross-sectional area, averaged volumetric flow rate, and mean pressure, respectively. The equations system is closed with a relationship between the pressure *P* and area *A* [8,9]:(3)$$P-P_{0}=\frac{Eh_{0}}{r_{0}\left(1-\sigma^{2}\right)}\left(\sqrt{\frac{A}{A_{0}}}-1\right)$$

$A_{0}$ and $h_{0}$ represent the cross-sectional area and wall thickness at the reference configuration, respectively, and $P_{0}$ represents the reference pressure. *E* is the Young’s modulus, $r_{0}$ is the radius corresponding to $A_{0}$, and σ means the Poisson ratio. Based on the availability of patient-specific radius information, the radius of an artery may be represented either as a time-variant linearized function derived from the literature data or directly from patient-specific measurements. In the instance provided in this work, the patient-specific vascular geometry is obtained from the vascular centerline generated in the arterial geometric modeling module, and the reference geometry data are prescribed based on the investigation of refs. [11,24].

The original 1D–0D computational model was further extended by incorporating the cerebral circulation to account for the change in hemodynamics introduced by endovascular treatments. Zhang et al. [17] proposed the inclusion of patient-specific measurement data to apply the model to individual patient cases. Under the assumption that the sole immediate change following surgery is the removal of stenosis, Zhang developed a predictive method for assessing the risk of cerebral hyperperfusion syndrome (CHS) utilizing the 1D–0D simulation system.

## 4. Architecture

We developed an integrated system that fulfills the requirements outlined in Section 2. This system comprises four key modules as shown in Figure 2: (1) medical image processing, (2) arterial geometry modeling, (3) simulation, and (4) 3D visualization. The system’s graphical user interface (GUI) is built using the Microsoft Foundation Class framework, with core functionalities programmed in Visual C++ 2019 and a Fortran95 external simulation solver.

### 4.1. Medical Image Processing

The medical image processing module is designed for DICOM file input/output (I/O) and advanced image processing. To achieve this, the module integrates several specialized libraries and platforms, each contributing to different aspects of image processing workflow. The ITK library is utilized for reading DICOM files, which include a three-dimensional image matrix and header information to create an appropriate workspace. The Mist library is employed for image segmentation, extracting vessel pixels to construct vessel volumes. Additionally, the V-modeler platform [25] facilitates interactive image segmentation. For more efficiency, we also offer pre-trained convolutional neural network models for the automatic segmentation of major arterial structures [26].

### 4.2. Arterial Geometry Modeling

The arterial geometry modeling module processes vascular centerlines or skeleton models of vasculatures, as shown in Figure 3c. Centerlines can either be generated by thinning processing the vasculature volume of the segmented medical image or by importing a texture-formed node array that is stored in external files. Utilizing the Mist library, this module converts the three-dimensional vasculature volume into one-pixel lines by shrinking the surrounding pixels to voxelized skeleton. The voxelized skeleton can be represented as a series of nodes $\vec{N}$:(4)$$\vec{N}=\left(n_{0},n_{1},n_{2},\ldots ,n_{i},\ldots \right)$$
where the node $n_{i}$ contains three-dimensional coordinates and the local radius of vessel segments:(5)$$n_{i}=\left(x_{i},y_{i},z_{i},r_{i}\right)$$

The nodes are classified into three types by the connection status of the voxelized skeleton: terminal, bifurcation, and plain. For centerlines with bifurcation nodes, a pruning process is applied to remove extraneous branches, enabling users to focus computations on target areas and reduce noise from segmentation. The skeleton, once divided by bifurcations, is cut into segments and labeled by the arterial index.

Quantization noise in medical images can lead to local fluctuations, potentially impacting the accuracy of vessel radius and length estimations used in patient-specific geometry. This, in turn, affects both simulation results and the visualization of blood vessel surfaces. To mitigate these issues, the arterial geometry modeling module facilitates spline fitting on voxelized skeleton, resulting in a smooth and noise-resistant skeleton model. We employ the geo-SFM method [27] for spline fitting, which utilizes a 5th-degree spline basis function, 3rd and 4th derivative penalty terms, and employs the Akaike information criterion (AIC) for optimizing penalty term coefficients. This approach effectively minimizes curvature and torsion errors in the centerlines, after arterial geometry modeling, the centerline is separated as a set of nodes $\vec{P}$ with a grid width Δl as 2 mm:(6)$$\vec{P}=\left(p_{0},p_{1},p_{2},\ldots ,p_{i},\ldots \right)$$
where the node $p_{i}$ contains three-dimensional coordinates and the local radius:(7)$$p_{i}=\left(x_{i},y_{i},z_{i},r_{i}\right)$$

### 4.3. Simulation

Based on the 1D–0D blood flow model described in Section 3, the following variables can be computed:Flow Rate Q(s,t): This is the velocity of blood through the vasculature over time, quantifying the volumetric flow rate along the length of the vessel, where *s* represents the spatial coordinate along the vessel’s length, and *t* denotes time.Blood Pressure P(s,t): This represents the pressure exerted by the blood against the vascular walls over time, providing insight into the forces that the heart must overcome to circulate blood and the potential stresses on vessel walls.Cross-sectional Area A(s,t): This is the area of the blood vessel cross-section at different spatial locations and time points, reflecting changes in vessel diameter due to pressure dynamics and the mechanical properties of the vessel wall.

Additionally, wall shear stress (WSS, $\tau_{w_{i}}$) can be calculated through a post-processing step based on extra assumptions regarding the shape of the vessel.

Each vascular segment in this model corresponds to an index. Here we provide an example where the 1D component is constructed with a patient-specific cerebral arterial network. The 1D cerebral arterial network includes the circle of Willis, which contains 14 arteries. The network is divided into 20 arterial segments at bifurcation points. Figure 3a shows the structure of a 1D vessel segment with a+1 nodes. In this sample, the distance between adjacent nodes (Δs) is 1 mm. Each node *s* (s=0,1,2,…a) holds the cross-sectional area (A(s,t)), blood pressure (P(s,t)) and blood flow rate (Q(s,t)). As illustrated in Figure 3c, nodes in the 1D model ($(\vec{S})=\left(s_{0},s_{1},s_{2},\ldots \right)$) and those in the 3D model ($\vec{P}=\left(p_{0},p_{1},p_{2},\ldots \right)$, where $p_{i}=\left(x_{i},y_{i},z_{i}\right)$), are based on the same vessel index. This correspondence is crucial for accurately mapping simulation results onto the 3D spatial geometry. The data for each node, including A(s,t), P(s,t), and Q(s,t), are transformed into 3D coordinates as A(x,y,z,t), P(x,y,z,t), and Q(x,y,z,t).

In addition to the 1D–0D blood flow simulation results, the module provides a post-processing function to calculate the wall shear stress (WSS). This calculation is based on the cross-sectional area of the vessel and the velocity distribution across the vessel’s cross-section. The average blood flow velocity at the cross-section of $n_{i}$ is determined as follows:(8)$$V_{i}=\frac{Q_{i}}{A_{i}}$$

Assuming the vessel segment is a cylindrical tube, the velocity distribution $U_{k_{i}}$ of the cross-section can be obtained by the distance to the center *r*, according to the Hagen–Poiseuille law:(9)$$U_{k_{i}}=2V_{i}\left(1-\frac{r_{k_{i}}^{2}}{R_{i}^{2}}\right)$$
where $R_{i}$ is the radius of the cross-section at *i*-th node.

The wall shear stress $\tau_{w_{i}}$ at the cross section of node $n_{i}$ can be calculated as:(10)$$\tau_{w_{i}}=\tau_{k_{i}}{|}_{r_{k_{i}}=R_{i}}=4\mu V_{i}\frac{1}{R_{i}}$$

### 4.4. 3D Visualization

Visualization of computational results is achieved through rendering. This method maps 1D numerical results onto 3D surfaces. Although more complex approaches can be employed to map onto refined surfaces created by segmentation, considering the simplicity of the 1D computational results, we utilize a compact method to quickly display the 1D results and flow direction within vessel segments. This is accomplished by reconstructing the vessel lumen around the centerline, thereby facilitating a rapid representation of the physiological parameters of interest in the context of the vessel segment.

The reconstruction of the vessel lumens and the display of numerical results are achieved through color contours mapped onto a triangular surface mesh. The vessel lumens are rendered using triangle meshes by OpenGL. Each vessel segment presents the cross-section area computed by 1D–0D simulation. The procedure for generating this surface mesh is described below. To each node $P_{i}$ in the array $\vec{P}$, the right-handed coordinate system which holds the unit vector $\vec{nx_{i}}$, $\vec{ny_{i}}$ can be determined as:(11)$$\vec{dir_{i}}⊥\vec{nx_{i}},\vec{dir_{i}}⊥\vec{ny_{i}}$$
where $\vec{dir_{i}}$ represents the direction vector of node $P_{i}$. With provided the circular cross-section, the vessel radius upon node $P_{i}$ can be computed by $A_{i}$, the cross-section area of node $P_{i}$:(12)$$r_{i}=\frac{\sqrt{A_{i}}}{\pi }$$

Furthermore, the cross-section can be divided by *k* vertex $Q_{i,0},Q_{i,1},\ldots ,Q_{i,l},\ldots ,Q_{i,k-1}$ upon the circle. The vector $\vec{P_{i}Q_{i,l}}$ can be computed as:(13)$$\vec{P_{i}Q_{i,l}}=\left(\vec{nx_{i}}\cos\frac{2\pi }{k}+\vec{ny_{i}}\sin\frac{2\pi }{k}\right)r_{i}$$

Therefore, the corrdinate of dividing vertex $Q_{i,l}$ can be computed by $P_{i}$ and $\vec{P_{i}Q_{i,l}}$:(14)$$\vec{Q_{i,l}}=P_{i}+\vec{P_{i}Q_{i,l}}$$

For each node $P_{i}$, the vertices of each triangle mesh are determined by the dividing vertex $Q_{i,l}$, $Q_{i,l+1}$, and $Q_{i+1,m}$ or $Q_{i-1,m}$, the latter being the dividing vertex closest in Euclidean distance to an adjacent node. Specifically, for l=k−1, the triangle mesh is determined by $Q_{i,k-1}$, $Q_{i,0}$, and the nearest dividing vertex subordinate to an adjacent node. The schematic diagram of the triangle mesh generation is shown in Figure 4, where the emphasized red triangle is determined by $Q_{i,l}$, $Q_{i,l+1}$, and $Q_{i+1,m}$.

## 5. Results

In this section, we introduce practical applications of our proposed platform with a focus on data processing and visualization. It comprises two parts: (1) geometry modeling and (2) dynamic numerical visualization. Geometry generation demonstrates the visualization of centerline smoothing results using the SFM, applied to patient-specific arterial vascular networks derived from medical imaging data. Dynamic numerical visualization illustrates how the simulation result for the patient-specific case is visualized. We will highlight two cases to demonstrate the usability of the platform. The first case, Carotid Stenosis (CS), involves a detailed study of severe carotid stenosis. The second case, Vascular Remodeling in Celiac Artery (VRCA), focuses on the simulation of vascular diameter changes during celiac artery stenosis.

### 5.1. Case CS: Post-Surgical Flow Rate Prediction

We visualize the simulation results using the method proposed in Section 4. The vessel lumen‘s tubular surface is constructed based on the cross-sectional data computed in the simulation, with hemodynamic status depicted through color coding. The results are shown in Figure 5. Within each sub-figure, the left plot shows the pre-surgical result, and the right shows a post-surgical predictive result. Each sub-figure in the visualization presents distinct perspectives of the cerebral vasculature: Figure 5a displays a frontal view, Figure 5b shows a lateral view, Figure 5c shows a close-up of the Circle of Willis (CoW) and the sphenoid bone beneath it, and Figure 5d presents a detailed view of the CoW without the bone structures.

The CoW is a ring-shaped arterial network, connecting the anterior, middle, and posterior cerebral arteries along with communicating arteries. Due to this unique structure, a decrease in flow rates caused by stenosis-induced pressure drops can lead to reversed flow. In a healthy network, the flow within the CoW is symmetrically distributed between the left and right sides. Blood flow from each Internal Carotid Artery (ICA) diverges at the bifurcations of the Middle Cerebral Artery (MCA) and Anterior Cerebral Artery (ACA). In the case of the patient with stenosis, an imbalanced flow distribution in the CoW can be observed. A compensatory flow from the right to the left caused a flow reversal in the left ACA. This is evident in the pre-surgical results of Figure 5a,b, where an asymmetrical color distribution in the two large frontal arteries (ICAs) is noticeable. Furthermore, in Figure 5c,d, the pre-surgical results show retrograde flow marked by negative flow rate.

A virtual surgery was simulated to completely remove the stenosis from the left ICA and the stenosis rate becomes 0. The predictive result is shown on the right side of each sub-figure. From the post-surgical predictive results shown in Figure 5a,b, the removal of stenosis leads to a balanced flow rate distribution between both sides. Similarly, in Figure 5c,d, the reversal of flow is no longer present. These outcomes suggest that the endovascular treatment was successful and safe, achieving the desired effect of restoring symmetrical blood flow and eliminating abnormal flow patterns in the patient’s cerebral vasculature.

### 5.2. Case VRCA: Vascular Diameter Remodeling

This case involves a simulation of vascular remodeling during celiac artery stenosis. Celiac artery (CA) is the first major branch of the abdominal aorta(AA), which supplies blood to the liver, spleen, and stomach. In the presence of celiac artery stenosis, a collateral pathway between the common hepatic artery (CHA) and superior mesenteric artery (SMA) can undergo significant dilatation to maintain the blood supply to the organs, which is referred to as arterial remodeling. However, the collateral pathway from SMA might be removed in surgeries such as pancreaticoduodenectomy (PD). In that case, celiac artery stenosis can cause postoperative visceral ischemia [28]. Research has verified that arteries regulate their cross-sectional area to maintain constant wall shear stress while flow rate changes [29,30,31,32].

In the VRCA case, one of the arterial remodeling cases introduced in [19] is visualized. Three types of conditions are used to perform the simulation: normal (0% stenosis), 90% stenosis without remodeling, and 90% stenosis with remodeling. Figure 6 shows the time-average WSS (TAWSS) distribution of the AA, CA, SMA and a representative PD. Figure 6a shows the normal TAWSS distribution, Figure 6b shows that after the stenosis is attached, and Figure 6c shows the remodeled status. By comparing the first and the second image, it can be observed that with the introduced CA stenosis, PD is subjected to high TAWSS, which is more significant than adjacent arteries. After remodeling, the vessel radii of CA and PD are visibly enlarged in the third image; meanwhile, the TAWSS distribution is restored to the same level as the normal one. In this way, by visualizing the simulation results, the situation where the blood flow is compensated by the WSS adjustment of the PD in the CA stenosis case is reproduced.

## 6. Discussion

The flow regulation mechanism in the vascular network is a point of interest within the research of blood flow numerical simulation. Simulating three-dimensional looping structures such as cerebrovasculatures usually requires enormous computing resources (e.g., by supercomputers [33]), which is usually unimaginable when handling case studies.

In response, 1D simulation has emerged as a type of ROM developed to provide a cost-effective yet reasonably accurate alternative to full-scale 3D simulations. Reymond et al. [34] categorized and compared 1D blood flow models by several criteria, including the inclusion of systemic circulation, the integration of heart models, and consideration of arterial wall viscoelasticity. The model utilized in this study is a 1D–0D systemic blood flow model that incorporates a heart model and a cerebral arterial tree. The 1D–0D model leverages patient-specific data to adjust distal vasculature models, offering a tailored approach to simulating physiological blood flow dynamics with enhanced relevance to individual patient conditions. The model can be applied to study the global hemodynamic influences of stenoses located in various regions.

We developed a 1D–0D integrated system with a 3D visualization function. The system consists of four modules to cover the image-based simulation workflow: medical image, data pre-processing, simulation/post-processing, and visualization. The medical image module is based on the modeling application V-modeler. This module performs medical image processing, allowing users to create 3D vascular volume and extract basic vascular centerlines. The arterial geometry module is used to generate proper 1D model input for simulation based on the basic vascular centerlines. The system is equipped with a spline-fitting function to restore the geometric characteristics of blood vessels and reduce the possible impact of quantization noise of medical images on blood flow simulation. For the simulation module, the 1D–0D solver based on the work of refs. [15,16] and improved by refs. [17,19] is employed, which is optimized for patient-specific data to predict the blood flow in the arterial network with circulation before and after the stenosis is removed. The output of the solver contains cross-sectional area (A), blood pressure (P), and blood flow rate (Q). WSS can be obtained by the basic output through a post-processing function. The last module is the three-dimensional dynamic visualization module. It uses the cross-sectional area solved by the simulation module to compute the time-varying vessel radius to construct a triangle mesh vessel lumen around the arterial centerlines. The hemodynamic information such as flow rates and blood pressure solved by the simulation module is projected on the vessel lumen to generate dynamic visualization.

Severe carotid artery stenosis poses a risk of cerebral infarction and stroke, necessitating endovascular interventions like Carotid Artery Stenting (CAS) or Carotid Endarterectomy (CEA). However, postoperative complications such as cerebral hyperperfusion syndrome (CHS) can lead to cerebral hemorrhage. Rapid and accurate surgical planning is crucial for the safety of these procedures. In this context, ROM offers an appropriate choice for cerebral blood flow simulation due to their flexibility and low computational cost. The case study presented in this paper involves a patient with significant stenosis (70% NASCET). The example demonstrates medical image-based modeling of the patient’s cerebral vasculature and patient-specific preoperative blood flow computation and postoperative prediction. The results, presented through a visualization module, provide insightful outcomes and essential numerical information, underscoring the utility of ROMs in facilitating safe and effective surgical planning.

The VRCA case study underscores the pivotal role of visualization in deciphering complex physiological processes, particularly arterial remodeling in celiac artery stenosis scenarios. This study demonstrates the utility of visual tools in representing vascular diameter variations and Wall Shear Stress (WSS) dynamics under diverse conditions, including normal, stenotic, and post-remodeling states. A critical observation from the VRCA case is the detailed portrayal of the vascular system’s adaptive response to stenosis. The visualization illustrates the compensatory dilation of specific vessels in response to reduced blood flow, a mechanism vital for maintaining organ perfusion. Incorporating user interaction into this visualization framework enhances the utility of the system, facilitating a comprehensive understanding of these complex processes and enabling hypothetical scenario analyses.

Despite the presentation of a limited set of examples focused on stenosis, our platform maintains minimal assumptions regarding data format processing, thereby offering a measure of flexibility. This attribute renders the platform suitable for applying to vascular networks that can be effectively simplified into 1D representations. Nonetheless, this study is subject to certain constraints: First, the estimation of WSS is contingent upon the presumption that vessel segments are cylindrical and that blood flow conforms to the Hagen–Poiseuille law, implying an assumption of laminar and axisymmetric flow. Such a presumption might not hold across various physiological conditions, particularly at sites of vascular bends or bifurcations. Second, the technique of projecting 1D numerical outcomes onto reconstructed 3D surfaces enhances the model’s conciseness and versatility but could potentially compromise the visual authenticity of the depicted vascular dynamics.

## Figures and Tables

**Figure 1 bioengineering-11-00313-f001:**
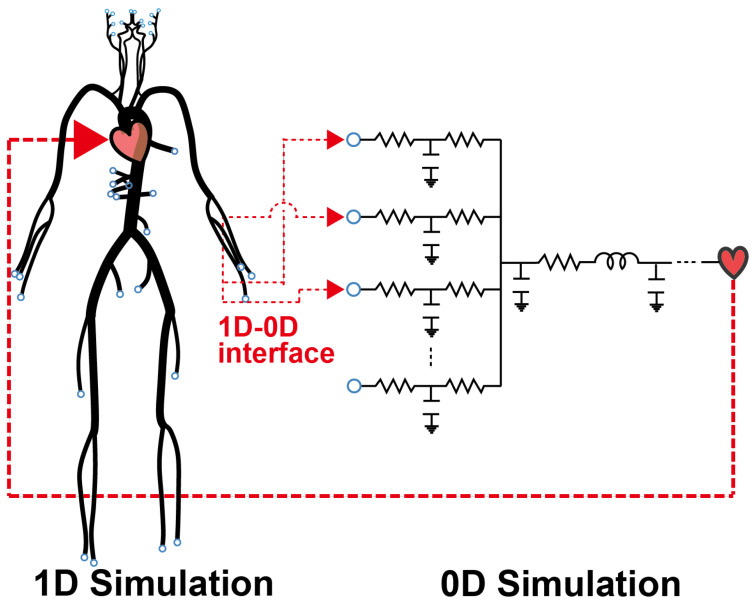
1D–0D simulation model.

**Figure 2 bioengineering-11-00313-f002:**
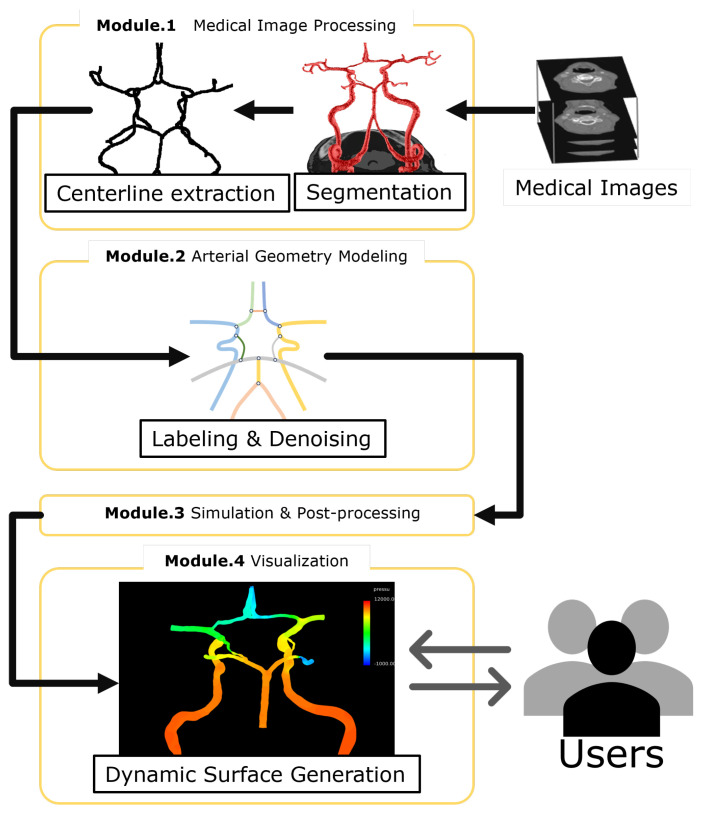
Four modules that form the proposed integrated system.

**Figure 3 bioengineering-11-00313-f003:**
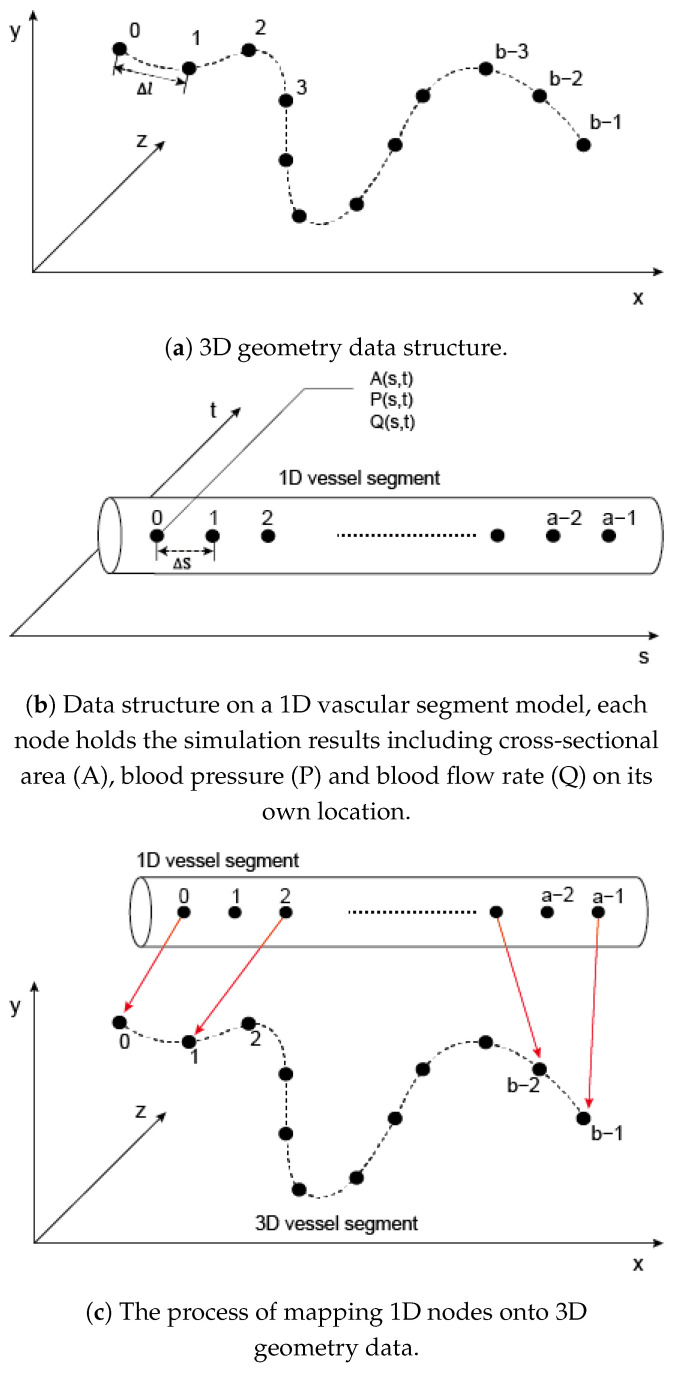
1D and 3D data structures and the mapping process.

**Figure 4 bioengineering-11-00313-f004:**
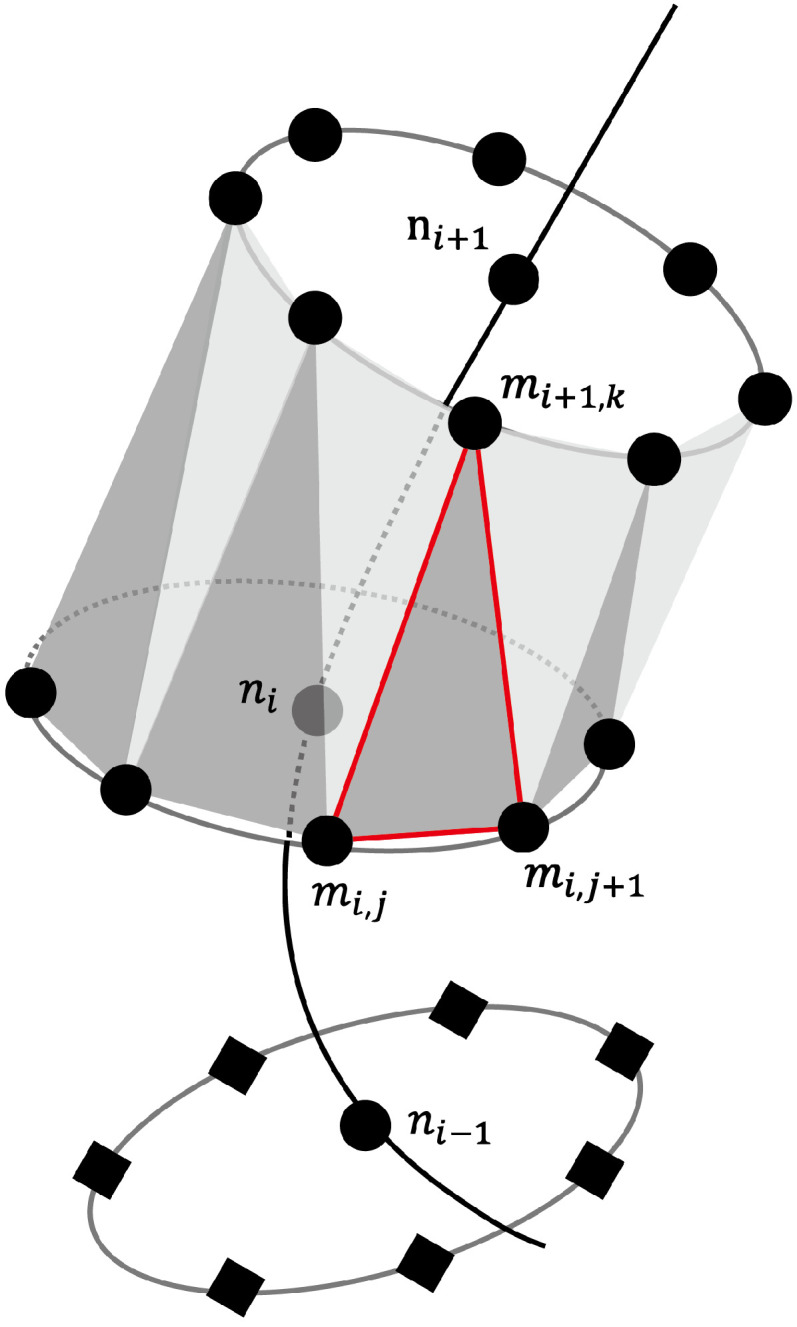
Triangular mesh generation diagram. Red lines highlight an example triangle.

**Figure 5 bioengineering-11-00313-f005:**
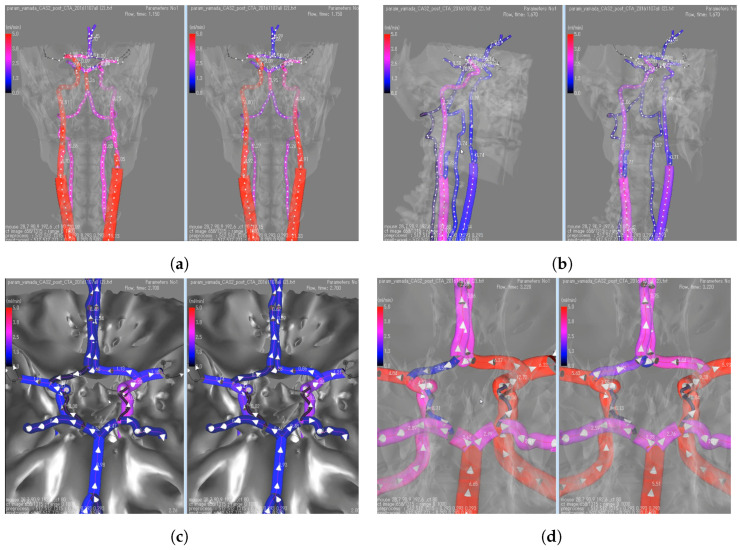
Dynamic visualization of blood flow rate in the CoW. In each sub-figure, images on the left show the diseased status and images on the right show the post-operation status.

**Figure 6 bioengineering-11-00313-f006:**
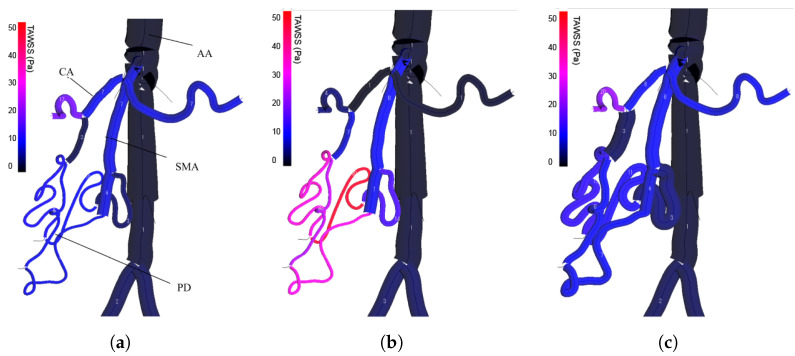
Dynamic visualization of TAWSS of the VRCA case.

**Table 1 bioengineering-11-00313-t001:** Comparison of 3D Models, ROMs, and Machine Learning Methods in Blood Flow Simulations.

Aspect	3D Models	ROMs	Machine Learning Methods
Complexity	High; detailed spatial resolution	Lower; simplifies geometry and dynamics	Varies; algorithm and data-dependent
Computational Cost	High; requires more computational resources	Lower; more efficient due to simplifications	High for training; data acquisition costly
Result Fidelity	High; captures detailed flow dynamics	Lower due to simplifications	Varies; dependent on training quality
Flexibility	Less flexible; hard to adapt to new cases	More flexible; easier to adapt	Adaptable, but may require retraining
Data Requirement	Geometry and flow specifics	Simplified vessel and flow characteristics	Extensive data for training
Application	Detailed analysis and research	Quick assessments, preliminary studies	Predictive modeling, pattern recognition

## Data Availability

Due to the sensitive nature of the medical data used in this study as well as considering the proprietary elements of the software developed in collaboration with corporate partners, the data presented in this study are available only upon request. Requests for access to the data can be directed to the corresponding author. The data are not publicly available to ensure the privacy of individuals and to adhere to legal and ethical standards.

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
