# Peer review of "Development of a 3D Vascular Network Visualization Platform for One-Dimensional Hemodynamic Simulation"

_bioengineering, 2024, doi:10.3390/bioengineering11040313_

Round 1

Reviewer 1 Report

Comments and Suggestions for Authors

Review report of the manuscript: bioengineering-2906348

by Y. Chen et al.

The authors provide a visualization tool for patient-specific and image- based 1D-0D simulations. The platform covers the entire workflow, from modeling to 3D simulation and visualization. The presented tool is intended to be helpful in clinics for surgical planning.

GENERAL COMMENTS

The presented platform is interesting, however, the main goal of the work in not clearly defined. Its application to two specific clinical cases does not necessarily prove the capacity of the tool to prevent complications in general.  I would hence tone down the capability of the method. On the contrary, in my opinion, the presented workflow can be considered a valuable tool capable of simulating and visualizing cerebrovascular blood flow in real time. As such, it is applicable specific cases such as those presented within the manuscript. And I guess to many other applications not described. 

The main flaw of the work is the absence of a comparison with the results of a 3D computational analysis that would assess the precision of the presented platform. Of course the methodology is much more rapid but surely lose some details.  

SPECIFIC COMMENTS:

-Is the platform flexible? Is it the platform suitable for visualizing diseased flow in other organs as an example?

-In the Introduction, it is mentioned the 'rapid decision making' in clinics. Howe about the precision of the methodology? The work has been not compared with a 3D computational study and visualization for assessing its validity. Of course, the presented methodology is much more rapid respect to a 3D analysis. However, an overall comparison is missing... 

-The Introduction section should include much more references of other tools developed and programmed recently for the blood flow in the cerebral arteries and capillaries. There are a considerable work in this field and it seems everything shuffled in the background of the presented platform. Please include all the studies that attempt similar topics of computing and visualizing 1D flow in the cerebral arteries. Some of these works also provide the possibility of introducing loops such as those of the Willis circle (see the works of Lorthois and coworkers and Linninger and coworkers as an example. there are many other as well).

- Is the 3D visualization a rendering? As far as I understood, the 3D visualization represents in the 3D vessels the intensity of the flow only, not the spatial distribution in a specific section...

-How many other variables can be computed with the methodology? Can WSS derived variables (important for the cerebrovascular hemodynamics) be obtained? Hematocrit distribution? Energy loss in the blood circuit?

-Please describe not only the capability of the model but also the limitations of the method and the limits of its applicability. 

-The Discussion should comment the results of the work within the existent literature. The authors mention only very few articles and they not really discuss their findings with the existent studies. Please include this. 

Reviewer 2 Report

Comments and Suggestions for Authors

The authors present an interesting computational model for the  3D vacsular visualization platform for one- dimensional hemodynamic simulation. Although I am not familiar with the mathematical background behind the model, the platform carries an important clinical significance in the fields of angiology, nephrology and vascular surgery. Therefore, with the reservation of a minor linguistic control, I propose the acceptance of the paper for publication.  

Comments on the Quality of English Language

Minor control required. 

Round 2

Reviewer 1 Report

Comments and Suggestions for Authors

The authors have responded to all the questions highlighted in the review report. I am pleased to recommend the manuscript for publication.